

# Water-use dynamics of an alien invaded riparian forest within
the mediterannean climate zone of the Western Cape, South
Africa
Scott-Shaw Bruce C[1], Everson Colin S[1,2,3] & Clulow Alistair D[4]
[1]Center for Water Resources Research, School of Agriculture, Earth & Environmental Sciences,
University of KwaZulu-Natal, Private Bag X01, Scottsville, Pietermaritzburg 3209, South Africa.
[2]South African Environmental Observation Network (SAEON), Grasslands-Wetlands-Forests Node, 1
Peter Brown Drive, Queen Elizabeth Park, Montrose, Pietermaritzburg 3201, South Africa
[3]Department of Plant and Soil Sciences, University of Pretoria, Private Bag X20, Hatfield, Pretoria
0028, South Africa.
[4]Agrometeorology, School of Agriculture, Earth & Environmental Sciences, University of KwaZulu-
Natal, Private Bag X01, Scottsville, Pietermaritzburg 3209, South Africa.
**Abstract**
In South Africa the invasion of riparian forests by alien trees has the potential to affect the
country's limited water resources. Tree water-use measurements have therefore become an
important component of recent hydrological studies. It is difficult for government initiatives,
such as the Working for Water (WfW) alien clearing programmes, to justify alien tree removal
and implement rehabilitation unless a known hydrological benefit can be seen. Consequently
water-use within a riparian forest along the Buffeljags river in the Western Cape of South
Africa was monitored over a three year period. The site consisted of an indigenous stand of
Western Cape afrotemperate forest adjacent to a large stand of introduced *Acacia mearnsii*.
The heat ratio method was used to measure the water-use of a selection of representative
indigenous species in the indigenous stand, a selection of *A. mearnsii* trees in the alien stand
and two clusters of indigenous species within the alien stand. The indigenous trees in the alien
stand at Buffeljags river showed significant intraspecific differences in the daily sap flow rates
varying from 15 to 32 L·day$^{-1}$ in summer (sap flow being directly proportional to tree size). In
winter (June) this was reduced to only 7 L·day$^{-1}$ when less energy was available to drive the
transpiration process. The water-use in the *A. mearnsii* trees showed peaks in transpiration
during the months of March 2012, September 2012 and February 2013. These periods
corresponded to favourable climatic conditions of high average temperatures, rainfall and high
daily vapour pressure deficits (VPD - average of 1.26 kPa). The average daily sap flow ranged
from 25 L to 35 L in summer and approximately 10 L in the winter. The combined accumulated
daily sap flow per year for the three *Vepris lanceolata* and three *A. mearnsii* trees was 5 700
and 9 200 L respectively, clearly demonstrating the higher water-use of the introduced *Acacia*
trees during the winter months. After spatially upscaling the findings, it was concluded that,
annually, the alien stand used nearly six times more water per unit area than the indigenous
stand. This finding indicates that there would be a hydrological gain if the alien species are
removed from riparian forests and rehabilitated back to their natural state.
*Key Words:    Sap flow, transpiration, indigenous trees, introduced trees, upscaling*





# 1 Introduction

While extensive research has been undertaken on the water-use of terrestrial ecosystems in South Africa, little is known about riparian tree water-use and growth of both indigenous and introduced tree species. This knowledge gap, as well as the poor ecological condition of South African riparian habitats, has led to uncertainty and contention over riparian rehabilitation techniques. The deep fertile soils, with high soil moisture contents associated with riparian areas, make them ideal for plant establishment and growth (Everson *et al*., 2007). As such, these areas are extremely vulnerable to invasion by pioneer plant species, particularly alien hybrid species that have historically been introduced for commercial forestry. It is widely believed that riparian zone vegetation, which can be described as the interface between terrestrial and aquatic ecosystems (Richardson *et al*., 2007), has a significant impact on the hydrology of a catchment due to the close proximity of riparian vegetation rooting systems to the water table. Most riparian trees are phreatophytic, meaning they have access to a permanent source of water because their rooting system is within the shallow ground water.

Through the process of evaporation and transpiration, riparian vegetation influences streamflow rates, ground water levels and local climates (Richardson *et al*., 2007). Vegetation along riverbanks filter surface and subsurface water moving across and through the soil to the river channel and therefore helps to maintain channel water quality, by regulating the water temperature (through shading), bank stability and turbidity and traps debris (Askey-Dorin *et al*. 1999). Riparian vegetation can access a wide range of water sources within the riparian zone, which includes rainfall, soil water, stream water and groundwater (O'Grady *et al*., 2005). Commerical forestry has been blamed for increasing the green water (water lost by total evaporation) and decreasing the blue water (water in rivers and dams) in areas across South Africa (Jewitt, 2006). There is a widespread belief in South Africa that indigenous tree species, in contrast to the introduced trees, are water efficient and should be planted more widely in land restoration programmes. This is based on observations that indigenous trees are generally slow growing, and that growth and water-use are broadly linked (Everson *et al*., 2008; Gush, 2011). However, tree water-use is technically difficult and expensive to measure, and so there is scant evidence of low water-use by indigenous trees.

For many field and modelling applications, accurate estimates of total evaporation (ET) are required, but are often lacking. Modelled estimates are often used without proper validation, and the verification of the results is questionable, especially in dynamic and highly-sensitive riparian areas. With the on-going development of micrometeorological techniques, it is possible to accurately quantify the various components of the water cycle over various terrestrial surfaces. The use of micrometeorological techniques is largely dependent on location, time constraints and available funds. However, due to continuous research by experts, the implementation of these techniques has become faster and more easily understood. In addition, comparisons between techniques and up-scaling have become possible, allowing for greater freedom in the choice of techniques and the length of measurement (Savage *et al*., 2004; Jarmain *et al*., 2008). Sap-flux density studies have been undertaken locally and internationally, and are well documented. Sap flux density measurements give precise information on flow directions and spatial flow distribution (Vandegehuchte and Steppe, 2013). The heat pulse velocity (HPV) method is the most accurate of the available methods when compared against gravimetric methods (Steppe *et al*., 2010; Vandegehuchte and Steppe, 2013).

The Buffeljags River site in the Western Cape has been an ecological research site since 2006 and forms part of a selective thinning experiment designed to assist Working for Water (WfW) clearing programmes. The government-funded WfW programme clears catchment areas of invasive alien plants





with the aim of restoring hydrological functioning while also providing poverty relief to local communities (Turpie *et al*., 2008). The aim of this study was to measure tree water-use to quantify the potential hydrological benefit of these forest management practices, improve the realistic modelling of these management approaches and provide guidelines towards suitable indigenous alternatives.

## 1.1    The study area

The Buffeljags river flows southwards along the Langeberg West mountain range into the Buffeljags dam. The Buffeljags river study area is at latitude 34˚00'15"S and longitude 20˚33'58"E (Figure 1), approximately 95-110 m above mean sea level. The research area is within Quaternary Catchment (QC) H70E and falls under the Western Cape Afrotemperate forest type which is characteristic of very small forest patches occuring along boulder screes consisting of streams, gorges and mountain slopes (Geldenhuys, 2010). The Langeberg Mountains consist of Table Mountain Sandstone/quartzite (north of the Buffeljags River) with a ridge of shales to the south of the river. The soils are characterised by structureless sands, a result of previous alluvial deposition.The climate is typical of the Western Cape with hot summers and cold winters. However, the rainfall is fairly evenly spread throughout the year. The long-term (137 year record) mean annual precipitation (MAP) at Buffeljags River is 636 mm. The daily maximum temperatures range from 17.1 °C in July to 27.5 °C in January. The mean daily minimum temperature is 15 °C in February and 5 °C in July. A 99 ha riparian forest occurrs along the river with 75 ha of invaded forest (lower reach) and 24 ha of pristine indigenous forest (upper reach) (Figure 1), comprising of species such as *Celtis africana*, *Vepris lanceolata*, *Prunus africana, Rapanea melanophloeos* and *Podocarpus falcatus***.** The stand height ranged from 3 to 15 m in the indigenous stand and 11 to 17 m in the alien stand. The surrounding vegetation is mountainous fynbos and renosterveld.

Historically *A. mearnsii* trees were planted for small scale uses (firewood and building material) on the nearby farms. Working for Water cleared most of the alien trees which have since grown back over the last 15 years. Currently the invasion extends approximately one kilometer along the river.

## 1.2    The study sites

Three representative trees within the indigenous stand were instrumented for monitoring sapflow. These trees included an understorey tree (*Rothmania capensis*), one medium (*Vepris lanceolata*) and one large evergreen tree (*Vepris lanceolata*) that were most common throughout the stand. The leaf area index (LAI) within this stand was 3.6 throughout most of the season with a slight reduction during the winter months. Downstream of this site, within the alien stand (Figure 1), three *A. mearnsii* trees were instrumented over the three year study period. In a similar way, small, medium and large diameter classes were chosen to assist in the up-scaling of single tree transpiration measurements of the *Acacia* trees. The LAI of the *Acacia* stand was 3.1 during the summer months and 2.8 during the winter months. Two indigenous tree clusters within the alien stand were also instrumented. The *Vepris lanceolata* cluster contained two medium and one large diameter class trees (LAI of 3.4) while the *Celtis africana* cluster contained two large diamter class trees with a LAI of 3.3 in the summer months and 1.8 during the winter months. The LAI provided an indication of the seasonality of the trees and the light variations between the sites.

Both the indigenous and introduced alien stands were in a climax state with most of the canopy trees falling into the medium or large size classes. Although there were many smaller trees (excluding trees with a Ø < 5 mm), these did not contribute significantly to the total transpiration as they were shaded




out by the climax trees. An overview of the individual tree characteristics has been provided in Table
2    1.

**2    Methods**
A meteorological station was established on the 25[th] of January 2012 at Buffeljags River in a nearby
planted *Eragrotis plana* field, 1.6 km from the indigenous site. Rainfall (TE525, Texas Electronics Inc.,
Dallas, Texas, USA) at a height of 1.2 m from the ground was measured with additional measurements
at a height of 2 m for air temperature and relative humidity (HMP45C, Vaisala Inc., Helsinki, Finland),
solar irradiance (LI-200, LI-COR, Lincoln, Nebraska, USA), net radiation (NR-Lite, Kipp & Zonen,
Delft, The Netherlands) windspeed and direction (Model 03002, R.M. Young, Traverse city, Michigan,
USA). These were measured at a 10 second interval and the appropriate statistical outputs were recorded
every hour.
A Heat Pulse Velocity (HPV) system using the heat ratio algorithm (Burgess, 2001) was set up to
monitor long-term sapflow on all of the selected trees over a three year period. The instrumentation is
described by Clulow *et al*. (2013) and included a 0.5 second heat source (sapflow trace) in the form of
a line heater. A pair of type T-thermocouple probes was used to measure pre- and post-temperatures 5
mm above (downstream) and below (upstream) of the heater probe (Clulow *et al*., 2013). Hourly
measurements (CR1000, Campbell Scientific Inc., Logan, Utah, USA) were captured over the three
year monitoring period (January 2012 to March 2015).

An assessment of the bark and sapwood depth was undertaken on the selected trees using an increment
borer. This assessment assisted in determining the HPV probe insertion depths and the calculation of
sapwood area. The heat pulse velocity ($V_h$) was calculated from:

$$V_h = \frac{k}{x} \ln\left(\frac{V_1}{V_2}\right) 3600 \qquad\qquad (1)$$

where, $k$ is the thermal diffusivity of green (fresh) wood, $x$ is the distance (5 mm) above and below the
heater (representing upstream and downstream), and $v_1$ and $v_2$ are increases in the downstream and
upstream temperatures (from initial average temperatures) respectively. A thermal diffusivity ($k$) of 2.5
$\times 10^{-3}$ cm$^2$ s$^{-1}$ (Marshall 1958) was used. Wounding or damaged xylem (non-functional) around the
thermocouples was accounted for using wound correction coefficients described by Swanson and
Whitfield (1981). Sap velocities were then calculated by accounting for wood density and sapwood
moisture content as described by Marshal (1958). Finally, sap velocities were converted to tree water-
use ($Q_{tree}$) or sap flow (L·hr$^{-1}$) by calculating the sum of the products of sap velocity and cross-sectional
area for individual symmetrical tree stems (Clulow *et al*., 2013).

Tree growth was recorded every two months throughout the monitoring period by measuring diameter
at breast height with a dendrometer and canopy height using a VL402 hypsometer (Haglöf, Sweden).
Riparian forests typically have a narrow canopy with limited reach, which excludes techniques such as
eddy covariance and scintillometry being used to support the up-scaling of point water-use
measurements to stand water-use values. Given the homogenous characteristics of the alien stand and
the dominance of *Vepris* and *Celtis* species within the indigenouis stand, a methodology was followed
based on recent up-scaling studies (Ford et al., 2007; Miller et al., 2007). In addition, detailed stem
density data were available for the site due to extensive ongoing ecological research (Atsame-Edda,
2014). Medoid (representative of the population) trees were selected for sap flow measurement. This





included the most commonly occurring alien and indigenous species (canopy and understorey) and a
range of size classes for each species. A species density analysis was undertaken (Ø > 50 mm) in
replicated 400 m² plots per site. A relationship between total tree water-use ($Q_{tree}$ – L·day$^{-1}$) and each
representative size and species class was identified. This allowed for the estimation of the stand water
flux ($Q_{stand}$) which was divided by the plot area (400 m²) in order to obtain comparative units between
the indigenous and alien stands (L·day$^{-1}$·ha$^{-1}$). These values were then accumulated to annual values so
that the effect of alien and indigenous (evergreen and deciduous) stands on the water balance could be
quantified throughout a hydrological year.
The *A. mearnsii* site had a thin litter layer consisting mostly of broken branches, bark and leaves
compared with the indigenous site which had a thicker litter layer with a large amount of organic matter
accumulated from the various tree species and understorey vegetation. Volumetric soil water contents
were measured hourly at both the indigenous and alien sites (concurrent to the HPV measurements)
with three time domain reflectometry (TDR) probes (Campbell Scientific. CS 615) installed
horizontally at each site at depths of 0.1 m, 0.3 m and 0.5 m. The TDR probes were connected to spare
channels on the CR1000 datalogger of the HPV system. With hourly volumetric water content
measurements, the response of trees to rainfall events, or stressed conditions, were monitored and
supported the interpretation of the HPV measurements. An observation borehole was installed at the
site to monitor the groundwater recharge as well as to confirm the assumption that all the trees within
the riparian forest had direct access to groundwater. Soil samples were taken to determine the
distribution of roots, bulk density and soil water content. These samples (taken at various depths
throughout the profile) were weighed before and after oven drying to determine the soil characteristics.
**3   Results**
**3.1   Weather conditions during the study period**
The MAP over the three year study was significantly higher than the long-term average (636 mm) by
300-500 mm (2012 to 2014 being 1017, 902 and 1127 mm respectively). The rainfall distribution was
variable (lacking a seasonal trend) throughout the three years with a mean monthly value of 85 mm
(Figure 2). There were numerous days of high hourly rainfall (to a maximum of 30mm·h$^{-1}$ & 102
mm·day$^{-1}$) demonstrating the prevalence of high intensity storms at the site (Figure 3). The solar
radiation peaked at 34 MJ·m$^{-2}$·day$^{-1}$ following the same seasonal trend to that of the daily minimum and
maximum temperatures.
The relative humidity (RH) ranged from 20 % to 90 %, with little seasonal trend. During periods of
high solar radiation, the atmospheric demand was high and correlated to peaks in transpiration rates.
An average daily temperature of 22.1 °C was recorded at Buffeljags River in the summer months.
During these months, daily maximum temperatures exceeded 40 °C. During the winter months, the
temperatures averaged 12.1 °C due to numerous days with low solar radiation, such as during rainfall
events and cloudy days, and would likely result in little to no transpiration occurring. The daily
reference total evaporation (ET$_o$) averaged approximately 1 mm in the winter period to 4 mm during
summer. The daily ET$_o$ peaked at 7.5 mm, which correlated to peaks in measured transpiration.
**3.2   Tree water-use**





For comparative purposes the water-use of similar sized *Vepris* and *Acacia* trees were compared during
the wet and the dry seasons (Figures 4 & 5). During the summer month of January, the *V. lanceolata*
tree water-use exhibited seasonal curves indicative of the clear sunny days and high correlation to the
solar radiation. The medium sized *V. lanceolata* (Ø 17.4 cm) used an average of 24 L·day$^{-1}$ during the
summer months and an average of 8 L·day$^{-1}$ during the winter months (Figure 4). The medium sized *A.*
*mearnsii* (Ø 16.7 cm) used an average of 10 L·day$^{-1}$ in the winter months, similar to that of the *V.*
*lanceolata*. In the summer months, the *A. mearnsii* used an average of 39 L·day$^{-1}$, significantly higher
than the indigenous tree (Figure 8). During significant rainfall periods (> 5 mm) there was little to no
water-use in both trees due to the low evaporative demand and the wet canopy.
Individual whole tree water use was significantly reduced in winter (May and June) for most of the
trees, dropping by approximately 75 %. This was attributed to fewer daylight hours in the winter months
which resulted in less available energy at this time of year to drive the transpiration process. From
November 2012 to March in 2013, all species showed a significant increase in water-use during this hot
summer period. The water-use in the *A. mearnsii* trees showed a distinct peak in transpiration during
the months of March 2012, September 2012 and February 2013. During March 2012, the high average
temperature (21.5 °C), a 76 mm rainfall event and high daily vapour pressure deficits (VPD) (average
of 1.26 kPa) contributed to a high atmospheric demand. On cloudless days with a high VPD and high
soil water, the trees would be expected to use more water. The average daily sap flow ranged from 15
L·day$^{-1}$ in the smaller class tree, 25 L·day$^{-1}$ in the medium class tree and 39 L·day$^{-1}$ in the large class
tree (Tables 1 & 2).
The daily summer water-use of two of the *V. lanceolata* trees (Table 2) in the upper indigenous stand
showed high water-use with an average of 19 L·day$^{-1}$ (medium class) and 37 L·day$^{-1}$ (large class). The
high water-use in the large tree was ascribed to its size and deep rooting system which is presumed to
have had easy access to ground water at this site due to the proximity to the river (10 m horizontal
distance). This was verified with the borehole levels and a root analysis at the site. The water level
ranged from 3.2 m to 4.8 m at the site where roots were observed to 5 m, while installing the borehole.
The water-use of the small understorey tree *R. capensis* had a much lower water-use (average of 8
L·day$^{-1}$) which indicated that although the understorey used less water, it still made a significant
contribution to the water balance given the abundance of understorey species in the indigenous forest.
The *Celtis africana* trees displayed a high water-use during the summer period. As this is a deciduous
tree, no water was used during leaf fall in winter. The largest *C. africana* tree had a canopy area of 75
m$^2$ and was the largest tree at the site. Approximately 37 700 L of water was transpired by this tree
annually during the measurement period (Table 2). Given that this species is deciduous, it is important
to note that this tree uses a high volume of water in summer when water resources are usually limited.
However, in a summer rainfall region, like eastern South Africa, this tree would not use water during
the low flow season when water resources are limited. This is important for management decisions
throughout rainfall zones in South Africa.
The indigenous cluster in the alien site had a LAI of 3.4, which was higher than the LAI of 3.1 under
the nearby *A. mearnsii* trees. The indigenous trees in the upper reach indigenous site had an LAI of 3.6.
Although the summer water-use was higher in the introduced trees, the radial sapwood area was larger
in the indigenous trees (up to 413 cm$^2$) than the introduced trees (up to 171 cm$^2$). Trees with the highest
sap velocities are therefore not necessarily those with highest whole tree water-use. However, this does
indicate that the alien trees are more effective users of water, relative to their sapwood area.





### 3.3 Soil profile and water content

The volumetric water content (VWC) in the alien stand at Buffeljags River was very low dropping to 7 % during dry periods (Figure 6). During high rainfall events the soil VWC exceeded 20%, showing a rapid but short response to rainfall. This indicates that the soil water moves through the soil profile rapidly with very little water being stored in the profiles, particularly in the lower profile. The soils had a dry bulk density ($\rho_b$) of 1.58 g.cm$^{-3}$, a particle density ($\rho_{particle}$) of 2.66 g.cm$^{-3}$ and a porosity ($\phi$) 0.42 %, typically characteristic of sandy soils. The drying curve, after an isolated event, took on average 22 hours from its peak to the lowest dry level (Figure 4). *Acacia* stands are known to have deep rooting systems, with observations of greater than 8 m in South Africa (Everson *et al*., 2006). This suggests that during dry periods, this stand can access water from deeper layers in the soil profile.

In the indigenous stand (Figure 7), the middle TDR probe (0.3 m) showed the highest water content. During the warmest period (December to April) there was very little water in the profile (even after rainfall events). This would suggest that the deeper roots from the indigenous species were readily using water below the TDR probe measurement depths as there was no correlation between transpiration and change in VWC. In contrast, the alien stand upper soil profile water content responded to rainfall events suggesting that interception storage (throughfall, stemflow and litter catch) played a significant role when comparing these stands. After an isolated rainfall event, the drying curve, of the soil profile at the indigenous site, took much longer (up to one week) from its peak to the driest level. The average water content was 5 %, lower than the alien stand, suggesting a difference in root activity given the same soil characterisics.

The VWC at both sites did not respond significantly to rainfall events under 5 mm unless during consecutive events. The average water table depth, measured using an observation borehole, ranged from 5.2 m below the ground surface during the dry season to 3.2 m below the ground surface during the wet season (excluding extreme events). The water table recharge time showed a strong relationship to the soil wetting and drying response time recorded at both sites. In conclusion, both the indigenous and introduced stands are energy limited rather than water limited as both had root contact with the water table.

### 3.4 Upscaling tree water-use

The results obtained from the research area were used to determine an actual annual water-use per unit area for both the invaded alien and pristine indigenous tree stands. Using the stem density per size class, stands of forest were compared rather than individual trees. The upscaled water use of the *A. mearnsii* stand was 5879 L ha$^{-1}$ for the small size class, 7639 L ha$^{-1}$ for the medium size class and 9981 L ha$^{-1}$ for the large size class. When upscaled for all species and size classes the total stand water use was approximately 5.85 ML·ha$^{-1}$·year$^{-1}$ (585 mm·yr$^{-1}$). This was 57 % of the average annual precipitation recorded during the monitoring period (1021 mm).

The annual water-use of the indigenous stand was 1209 L·ha$^{-1}$ for the small size class, 6321 L·ha$^{-1}$ for the medium size class and 18900 L·ha$^{-1}$ for the large size class. The upscaled indigenous stand used 1.01 ML·ha$^{-1}$·year$^{-1}$ (101 mm·yr$^{-1}$). Based on these results we concluded that the alien stand uses nearly six times more water per unit area annually than the indigenous stand. This roughly correlated to the growth rate of each stand, where the stem breast height diameter increase over the study period (recorded on each tree measured) was between three to eight times faster than similar sized indigenous trees.





The inter-species and size class water-use variations, particularly within the indigenous stand, highlight
the importance of good replications of a representative sample tree species and size classes. These
results also highlight that individual indigenous trees, such as the *C. africana*, can use more water than
an individual alien *A. mearnsii* tree. An example of this is the largest *Celtis* using 14 000 L more water
annually than the largest *A. mearnsii*. However, the *C. africana* tree had a much larger diameter and
had a large canopy area under which no other trees grew, whereas approximately ten medium sized *A.
mearnsii* trees could occupy the same area as this particular tree. The importance of upscaling using
representative samples of species and size classes is clearly demonstrated by the study.
**4    Discussion and Conclusion**
There is a widespread belief in South Africa that indigenous tree species, in contrast to introduced tree
species, use less water and should be planted more widely in land rehabilitation programmes (Olbrich
et al., 1996; Dye et al., 2001; Everson et al., 2007; Dye et al., 2008; Gush and Dye, 2008; Gush and
Dye, 2009; Gush and Dye, 2015). A review of relevant literature revealed a general paucity of
information relevant to both indigenous and introduced tree water-use, the methods of replication and
the techniques used. Internationally, improved HPV techniques have been used on various vegetation
types and the accuracy of these studies has been validated using gravimetric methods (Granier and
Loustau, 2001; Burgess *et al*., 2001; O'Grady *et* al., 2006; Steppe *et al*., 2010; Vandegehuchte and
Steppe, 2013; Uddin and Smith, 2014). In South Africa, the HPV technique has been shown to provide
accurate estimates of sap flow in both introduced tree species such as *A. mearnsii* and *Eucalyptus
grandis*, and indigenous tree species such as *Podocarpus henkelii* and *Celtis africana* (Smith & Allen,
1996; Dye *et al*., 2001; Everson *et al*, 2007; Dye *et al*., 2008). A key recommendation from the
literature, which has been emphasized in a recent study by Gush and Dye (2015), is that more indigenous
tree stand management research is needed in South Africa.
Spatial estimates of evaporation and transpiration are required but are difficult to obtain in remote areas
with limited reach. A large capital and human effort was invested towards this study in order to extend
the monitoring period, with a range of species and replicates. This allowed for an accurate comparison
of indigenous and introduced tree water-use. The Buffeljags River site is unique in that it is one of very
few sites within South Africa with an extensive rehabilitation programme that aims to assist WfW and
similar clearing programmes. The results showed that individual tree water-use varies depending on size
and species. Up-scaled comparisons showed that stem density is important to the accurate representation
of stand water-use. An introduced stand of *A. mearnsii* can use up to six times more water than a mixed
indigenous stand. This finding is significant in that it provides clear evidence to justify the highly
expensive clearing programmes, which have in the past lacked quantifiable data on the potential
hydrological benefits of alien plant clearing. The results also indicate that rehabilitation or clearing
programmes need to consider the seasonal rainfall variability of a site as planting of deciduous
indigenous trees may provide larger benefits in summer rainfall areas due to no transpiration during
periods when water resources are limited.
This study provides an ideal opportunity to validate remotely sensed ET data which could also be used
to identify spatial variations in vegetation water use. This future research will allow for the broader
extrapolation of alien plant water-use and benfits of clearing ripaian zones to similar areas outside of
the immediate study area. Results may be used to further validate transpiration simulations from
hydrological models, particularly in riparian areas.



*Acknowledgements.* The research presented in this paper forms part of an unsolicited research project
(Rehabilitation of alien invaded riparian zones and catchments using indigenous trees: an assessment
of indigenous tree water-use) that was initiated by the Water Research Commission (WRC) of South
Africa. The project was managed and funded by the WRC, with co-funding and support provided by
the Department of Economic Development, Tourism and Environmental Affairs (EDTEA). The land
owners, Brian and Janet Kilpen of Frog Mountain Inn, are acknowledged for allowing field work to be
conducted on their property. Assistance in the field by Dr Terry Everson, Matthew Becker and Liandra
Bertolli is much appreciated.



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





1    Table 1. Tree physiology and specific data required for the calculation of sap flow and up-scaling.

| Indigenous Forest site (upper reach) | Wood density ($m^3\ kg^{-1}$) | Moisture fraction | Average wounding (mm) | Diameter (mm) | Size Class (S/M/L) | Representative Stem Density (stems·$ha^{-1}$) |
|---|---|---|---|---|---|---|
| *Vepris lanceolata* | 0.66 | 0.42 | 3.4 | 199 | L | 24 |
| *Vepris lanceolata* | 0.63 | 0.42 | 3.7 | 134 | M | 65 |
| *Rothmania capensis* | 0.59 | 0.45 | 2.8 | 125 | S | 120 |
| Introduced/Alien Forest site (lower reach) | | | | | | |
| *Acacia mearnsii* | 0.54 | 0.89 | 3.2 | 121 | S | 650 |
| *Acacia mearnsii* | 0.73 | 0.47 | 3.2 | 167 | M | 200 |
| *Acacia mearnsii* | 0.61 | 0.71 | 3.0 | 194 | L | 50 |
| Indigenous Cluster (lower reach) | | | | | | |
| *Vepris lanceolata* | 0.66 | 0.45 | 3.2 | 166 | M | 65 |
| *Vepris lanceolata* | 0.65 | 0.45 | 3.2 | 174 | M | 65 |
| *Vepris lanceolata* | 0.66 | 0.47 | 2.9 | 202 | L | 24 |
| Indigenous Cluster (lower reach) | | | | | | |
| *Celtis africana* | 0.71 | 0.52 | 6.1 | 319 | L | 24 |
| *Celtis africana* | 0.71 | 0.50 | 6.0 | 422 | L | 24 |

2    *Note: The stem density was grouped as per size class



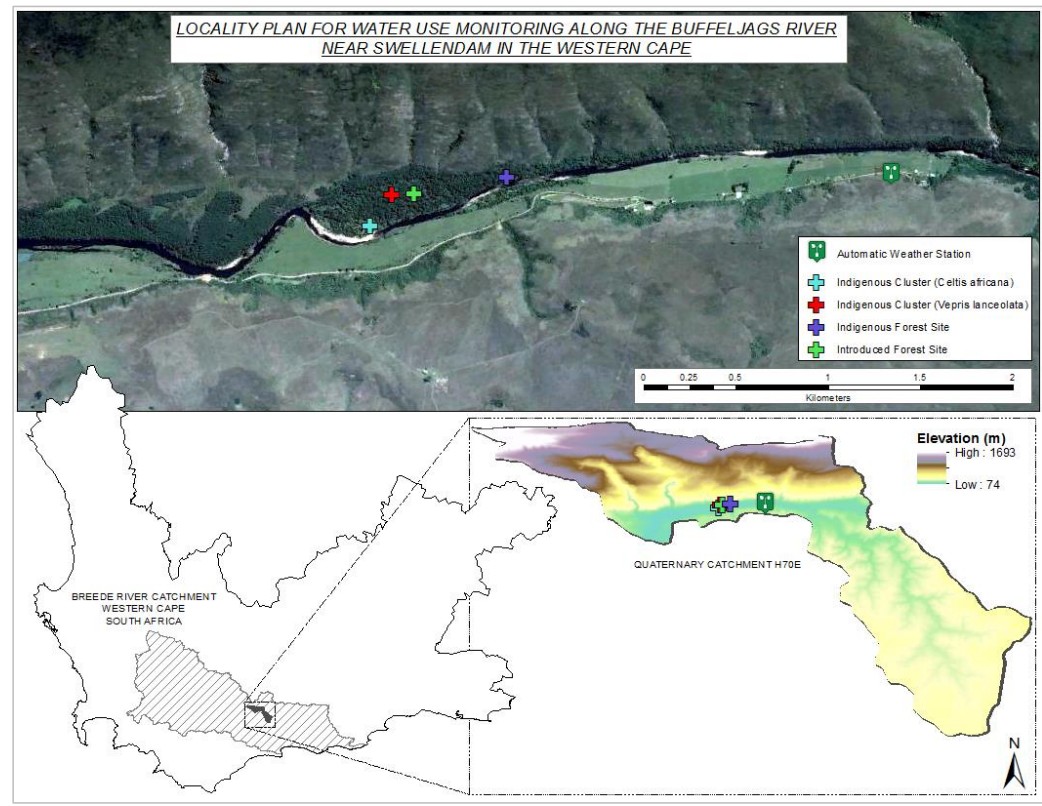

2  Figure 1.   Location of the Buffeljags River research area within the Western Cape, South Africa





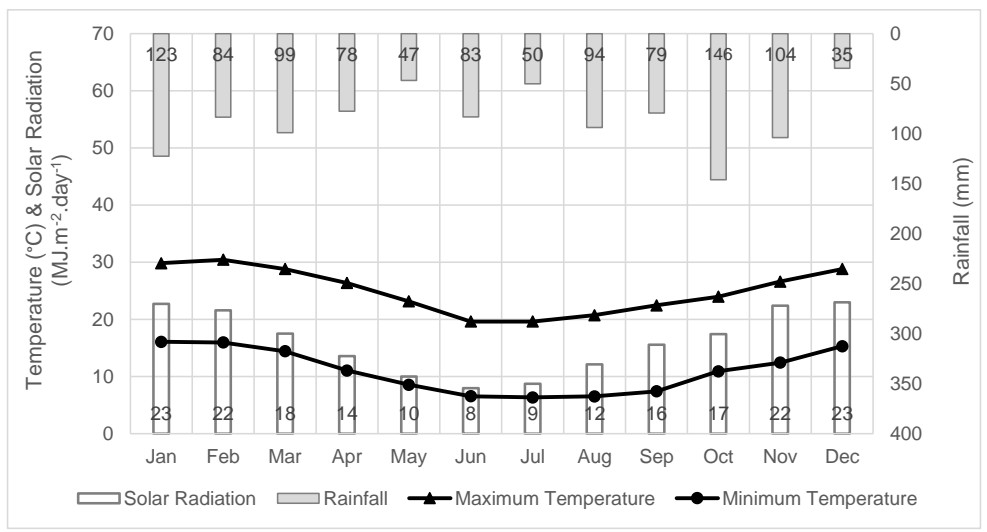

Figure 2.     The monthly rainfall, monthly solar radiant density, and average monthly maximum and minimum air temperatures at Buffeljags River averaged over three years.

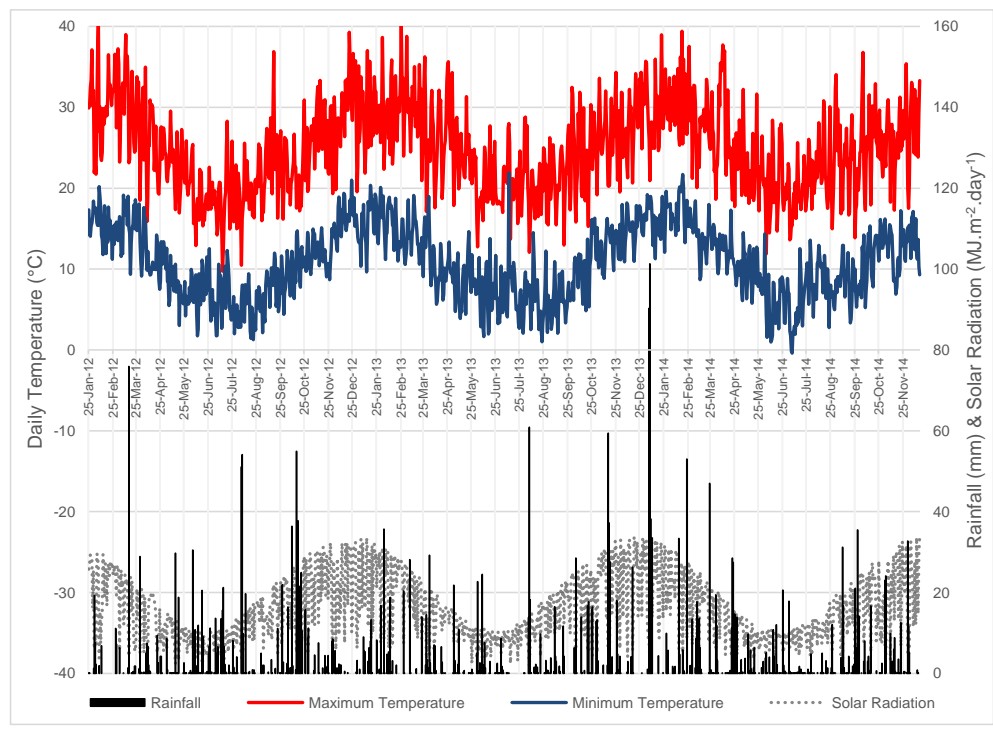

Figure 3.     The daily rainfall, solar radiant density and maximum and minimum air temperatures at Buffeljags River .





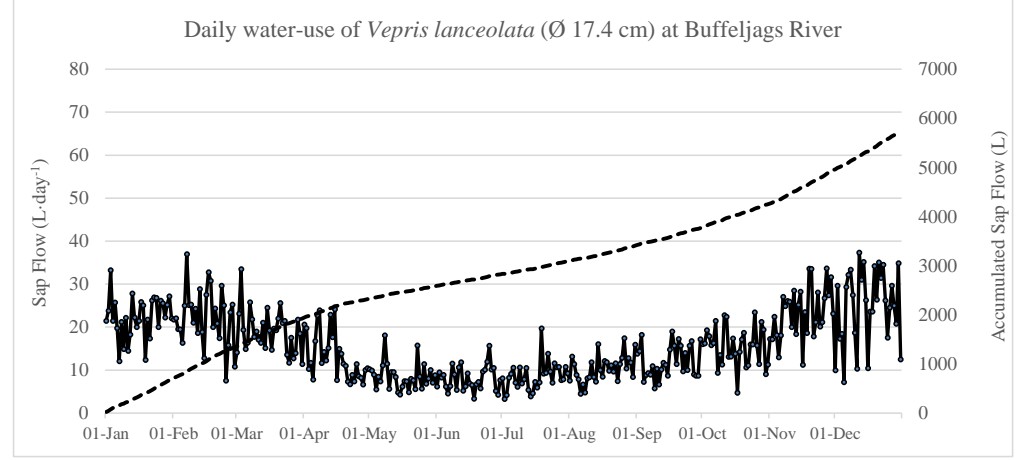

2  Figure 4.     Sap flow (daily and accumulated) from a *Vepris lanceolata* in the lower reach alien
3              stand at Buffeljags River (January 2012 to March 2015) averaged over three years

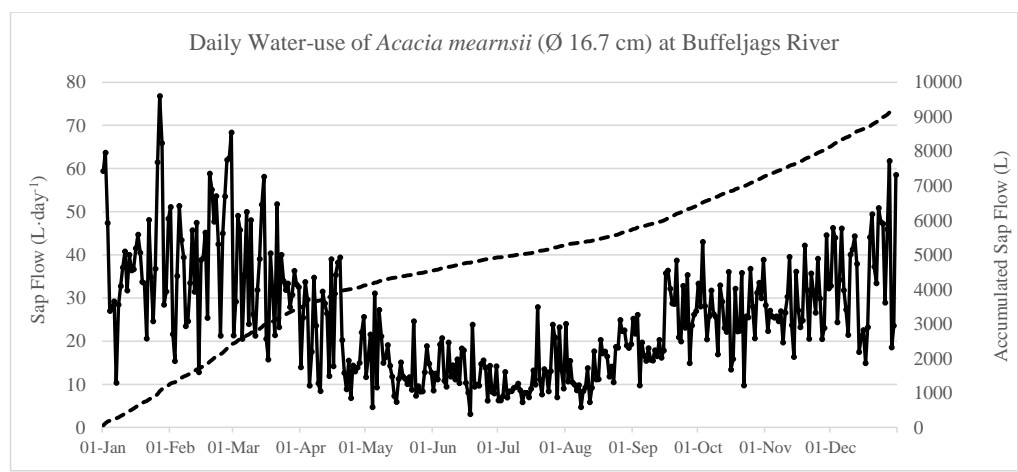

6  Figure 5.    Sap flow (daily and accumulated) from an *Acacia mearnsii* in the lower reach alien stand
7            at Buffeljags River (January 2012 to March 2015) averaged over three years
8




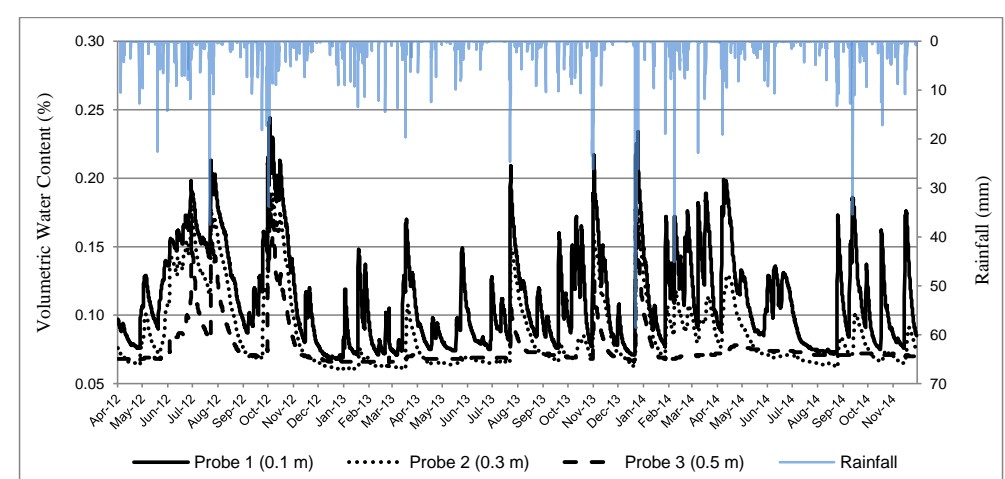

4   Figure 6.        Hourly volumetric water content of the lower alien stand corresponding to the hourly
5                          rainfall at Buffeljags River

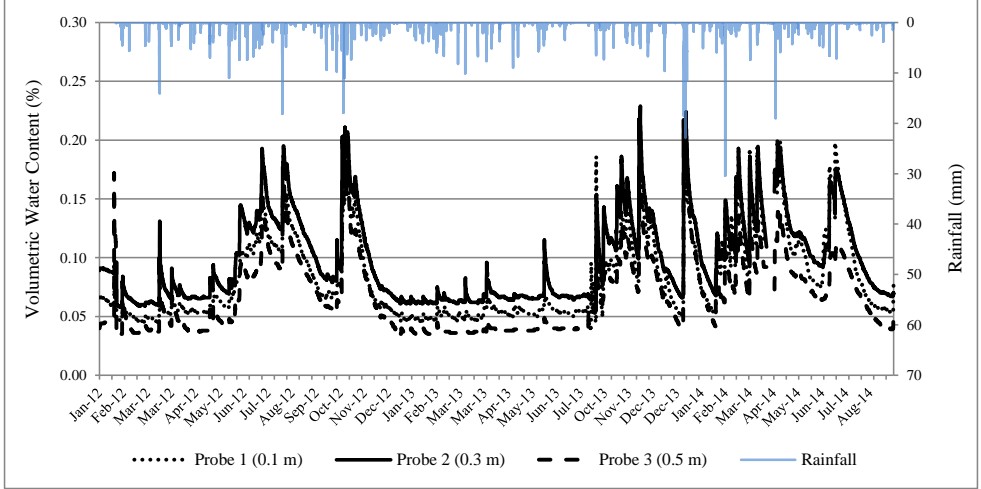

8   Figure 7.        Hourly volumetric water content of the upper indigenous stand corresponding to the
9                          hourly rainfall at Buffeljags River




1   Table 2.    Sap flow (daily and accumulated) for each species measured at Buffeljags River (January
2              2012 to March 2015)

| Forest Type / Location | Species | Daily Average Summer Sap Flow (L) | Daily Average Winter Sap Flow (L) | Annual Accumulated Sap Flow (L) |
|---|---|---|---|---|
| Indigenous Forest site (upper reach) | *Vepris lanceolata* | 19 | 7 | 6 534 |
| | *Vepris lanceolata* | 37 | 6 | 15 565 |
| | *Rothmania capensis* | 11 | 4 | 4 133 |
| Introduced/Alien Forest site (lower reach) | *Acacia mearnsii* | 25 | 8 | 9 226 |
| | *Acacia mearnsii* | 39 | 10 | 5 469 |
| | *Acacia mearnsii* | 32 | 9 | 7 207 |
| Indigenous Cluster (lower reach) | *Vepris lanceolata* | 14 | 6 | 5 725 |
| | *Vepris lanceolata* | 24 | 8 | 3 430 |
| | *Vepris lanceolata* | 39 | 14 | 9 174 |
| Indigenous Cluster (lower reach) | *Celtis africana* | 46 | 0 | 19 821 |
| | *Celtis africana* | 95 | 0 | 37 769 |

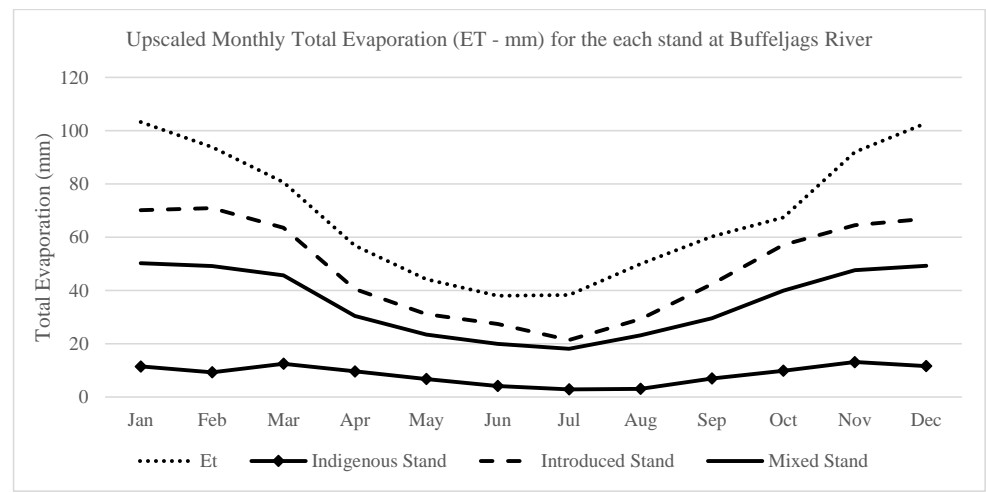

5   Figure 8.    Upscaled monthly total evaporation for the indigenous, introduced and mixed stands in
6              comparison to reference total evaporation
