# Peer review of "Water-use dynamics of an alien invaded riparian forest within the Mediterranean climate zone of the Western Cape, South Africa"

_Hydrology and Earth System Sciences, 2016_

## Referee Comment (RC1) · Anonymous Referee #1 · 20 Feb 2017

Manuscript Number: HESS-2016-650 (Scott-Shaw et al)

I have reviewed this manuscript and I have the following comments.

GENERAL COMMENTS The manuscript describes an experiment conducted in a riparian zone along the Buffeljags river in the Western Cape Province of South Africa. The authors quantified the transpiration dynamics of a mix of indigenous and invasive alien tree species in relation to climatic and soil factors. They observed significant differences in daily sap flow rates in both the indigenous and invaded stands. Overall, they conclude that the invasive alien trees used up to six times more water than the indigenous species. The authors assert that there would be a significant hydrological gain if the invasive alien species are removed from riparian forests and rehabilitated

back to their natural state.

EVALUATION

The manuscript is generally well written and it addresses a topical issue on the incremental water use by invasive alien plants over and above that used by indigenous vegetation. I have a few concerns though:

HESS is an international journal and my concern is that the way this manuscript is presented seems to target a South African audience. This is evident in the 'Introduction' and the 'Discussion and Conclusions' section where there is not a single reference to relevant international literature on alien invasive species. Where international literature is cited, it is on methodological issues, but not the invasive alien plants problem. Invasive alien plants are a global problem and the authors should present their manuscript in a way that appeals to an international audience. The authors have a rich data set which allows them to do this. For example, how do the water use rates of the invasive alien species studied here compare with that from studies outside the borders of South Africa? A more preferable route will be for the authors to use this data to validate a relevant international model e.g. MAESTRA, Shuttleworth and Wallace or others which other researchers elsewhere can use. The 'Discussion and conclusions' section is very weak. An extensive revision of that section is necessary along the lines suggested here and clearly stating what is novel about this study given that it is well known that alien species use more water than indigenous ones. Secondly, reference is made to total evaporation (evapotranspiration) in Fig. 8. Where was this measured or modelled? What assumptions are the authors making given that they only measured tree transpiration?

SPECIFIC COMMENTS - Abstract line 26: add the phrase . . . "of the heat pulse velocity sap flow technique" after the phrase . . ."heat ratio method". Not every one will understand this abbreviation. - Abstract general: presenting the water use values in liters only doesn't mean much to a water resources manager or other decision maker. I

would prefer to see area averaged transpiration values (in mm), compared for example, to ETo and/or rainfall. - Pg 3 lines 32-44: How was the LAI measured? - Pg 3 line 48: Indicate that the diameter was measured at breast height (DBH) - The HRM method of the HPV technique is well known in eco-physiological/eco-hydrological cycles. So there is no need for the details on Pg 4 lines 15 to 35. Remove this text. What would add value is more detail on: probe insertion depths and circumferential arrangements and how the conducting sap wood area was determined. Text for the water use modelling will also be appropriate in this section. - Pg 4 line 43 – sentence beginning with " Given the homogenous...." Doesn't make sense. Please revise. - Pg 5 line 20: To confirm direct use of ground water, it would have been nice to do this using stable isotopes rather. Include this data if you have it. - Pg 6 first paragraph: Fig 8 is discussed before Figs 6 & 7 which only appear on pg 7. Either revise the text or change the order of the Figs. There is also a mismatch with the units. The text, Pg 7 line 7 presents L/d, but Fig 8 presents water use in mm. Clarify this. - Pg 7, lines 39-40: Not clear if the annual transpiration and ETo are for a single year or averages over 3 years. Please clarify. - Pg 8 "Discussion and Conclusion". As stated earlier, this section is very weak and I would prefer to see; 1) comparisons between the results found here and what other researchers found, and; 2) comparisons with international literature. - I also have issues with the references: o The following references are missing in the reference list ïĆğ Everson et al., 2007; ïĆğ Richardson et al., 2007 ïĆğ O'Grady et al., 2005 ïĆğ Miller et al., 2007; ïĆğ Everson et al., 2006; ïĆğ Uddin and Smith, 2014. o Format of the references in the references list is not consistent. Some journals are written in full while others are abbreviated. In places it is not clear to who reports are written for e.g. pg 11 lines 10-11 and other places. For other references, journal names are replaced with address of the institution. For example, pg 11 line 38 is a publication in the Agricultural and Forest Meteorology journal which is bizarrely replaced by the institutional address. There are other such instances which need attention.

Comments end.

---

## Referee Comment (RC2) · Anonymous Referee #2 · 16 Mar 2017

General comments

This paper is a valuable contribution towards understanding differences in transpiration between indigenous and alien invasive tree species in South African riparian zones. The importance of this knowledge in justifying the huge Working for Water programme is highlighted. I feel that one or two additional references are worth mentioning in the introduction. The paper by Cullis, Gorgens and Marais (2005) is relevant to this paper, as it estimates just how seriously invasive trees in riparian zones can reduce stream-flows on a national scale. The papers by Scott et al (1994) and Scott and Lesch (1996) on the influence of trees in riparian zones within experimental catchments should also be referred to.

There are regular spelling, punctuation and grammatical errors throughout the paper. Please use a spell-checker and grammar checker in future to ensure that these do not get as far as the reviewers. We should be concentrating more on the content rather than pointing out these writing errors which are easy for the authors to detect and correct.

Specific comments

There was no reference to intermittent adjustments of probe depths to account for radial stem growth. Trees tend to grow past the probes, gradually positioning the TCs deeper in the sapwood, or even into the heartwood. I would expect that a species like A. mearnsii would grow substantially even over a period of 6 months or one year.

As is usual with sap flow studies, available equipment constrains the number of sample trees. Perhaps the authors would like to suggest how sampling intensity can be improved in future studies? If some of the trees were monitored for three years, maybe one could reduce this to one year, and then sample additional trees for each succeeding year?

I suggest that under "The study area", the mapped vegetation types be given, and a reference to Mucina and Rutherford added. Much more floristic and environmental information is then available to the reader.

The list of references needs some careful checking to ensure consistent formatting. 16 formatting errors picked up!

Technical corrections

P1, L26: Suggest using sap flow rather than water-use. Could be helpful since many readers will not know what the heat ratio method measures.

P1, L41: A hydrological gain could describe an increase in ET. The term is too general, better to say a gain in streamflow?
P2, L9: Hybrid genotypes are most common in Eucalyptus plantations, but they are mostly not invasive.

P2, L20. This sentence is far too long and clumsy.

P2, 42: Surely temporal flow patterns as well?

P2, L46: Buffeljags River and river seen in the text. Be consistent.

P3, L11: ...which is characterized by ....

P3, L19: spelling. occurs

P3, L22: Podocarpus changed to Afrocarpus?

P3, L34. Strictly speaking, Plant area index, since light interception is also by twigs, branches, stems.

P3, L40: Vepris. After first mention, abbreviate to V. lanceolata.

P3, L42. Spelling. diameter

P4, L8: insert comma

P4, L32. Check format of units throughout MS. Inconsistent.

P4, L40: ...dendrometer, and canopy height...

P4, L41: I think it would be clearer to refer to aerodynamic fetch rather than reach?

P4, L44: Spelling. indigenous

P4, L46: Is this stem density sph or the wood density of stems?

P4, L47: Not sure if it is worth introducing the term medoid to the readers!

P5, L6: Spelling again. Indigenous

P5, L19: Were observation borehole measurements occasional or continuous. Pity

that groundwater and root pattern data could not be included.

P5, L29. Spelling. significantly.

P5, L40: ...occasional maximum temperatures exceeded 40oC ?

P6, L33: After first mention, abbreviate to C. africana

P7, L3: ...very low, dropping ...

P7, L7: ..and a porosity of 0.42...

P7, L9: 22 hours to get to FC or wilting point? Seems very fast, even for a sand.

P7, L10: Spelling. greater

P7, L18: Throughfall and stem flow are surely not interception storage?

P7, L20: ...indigenous site took much longer ... No comma

P7, L38: ...water use... Be consistent about using a hyphen.

P8, L4: C. africana

P8, L28: aerodynamic fetch rather than reach?

P8, L34: ..a stand of introduced A. mearnsii... Sounds better than an introduced stand?

P8, L34: ..can annually use up to six ....

P8, L37: But see Cullin Gorgens and Marais paper which offers some quantified estimates.

P10, L1 onwards: Found 16 formatting errors! Please check carefully.

Table 1. Suggest be consistent in arranging trees from small to large. The first two groups of sample trees have a different sorting order. Moisture fraction is very variable in group 2 (A. mearnsii). Do you know why?

Figure 2. Numbers within bars along the X axis are partly obscured.

Figure 3. Caption says solar radiant density. Y2 axis label says solar radiation. Be consistent. Format of units again variable. The point separating each unit is sometimes in a lower or middle position. Check throughout the manuscript.

Figure 4: ...lower reach alien stand... Consider referring consistently to A. mearnsii rather than alien stands.

Figure 7. Pity no water table trends shown!

Table 2. Thousands separated by blank spaces here, but not consistently through the paper. Check throughout.

Figure 8. I think Et in the legend should be Et0 ? Again, introduced stand might be better described as A. mearnsii?

---

## Author Comment (AC1) · 28 Mar 2017

RC2: 'Water-use Dynamics of an Alien Invaded Riparian Forest within the Mediterranean Climate Zone of the Western Cape, South Africa', Anonymous Referee #2, 16 Mar 2017

HESS-2016-650

Anonymous referee #2 (AR2) is thanked for their thorough review. The thorough comments and suggestions provided were appreciated by the authors.

AR2 stated that the paper is missing some relevant literature citations. "The paper by Cullis, Gorgens and Marais (2005) is relevant to this paper, as it estimates just how

seriously invasive trees in riparian zones can reduce streamflows on a national scale. The papers by Scott et al (1994) and Scott and Lesch (1996) on the influence of trees in riparian zones within experimental catchments should also be referred to." • This comment was well received. All of the recommended papers were reviewed and references made in the paper. A more recent paper by Scott (1999) was used. Reference was made to the contrast of water use between riparian and upslope invaded areas. Some results were included on the reduction in streamflow yield if a riparian area were to be left uncleared.

2. AR2 stated that there were regular spelling errors throughout the paper. • The authors apologize for these errors. These have since been corrected.

3. There was no reference to intermittent adjustments of probe depths to account for radial stem growth. Trees tend to grow past the probes, gradually positioning the TCs deeper in the sapwood, or even into the heartwood. I would expect that a species. • The authors agreed with this comment. More detail was provided in the methods stating that regular checks were done to adjust probes for tree growth.

4. As is usual with sap flow studies, available equipment constrains the number of sample trees. Perhaps the authors would like to suggest how sampling intensity can be improved in future studies? If some of the trees were monitored for three years, maybe one could reduce this to one year, and then sample additional trees for each succeeding year? • This is a valid point. However, during the study the authors decided to focus on long term monitoring of the same trees that were all providing good clear data. This was also beneficial for investigations into the change of wounding over a longer period. For the indigenous stands, it was difficult to find representative trees within close proximity to one another for wiring to a central logger box. Had the equipment been moved, the authors would not have been able to measure a range of sizes elsewhere. In the introduced stand, there were few clusters of indigenous species limiting the possibility of moving the equipment. For consistency, the A. mearnsii equipment was not moved.

[Figure]

5. I suggest that under "The study area", the mapped vegetation types be given, and a reference to Mucina and Rutherford added. Much more floristic and environmental information is then available to the reader. • This information was included and reference was made to Mucina and Rutherford.

6. The list of references needs some careful checking to ensure consistent formatting. • This has since been checked and corrected.   Specific Comments

1. P1, L26: Suggest using sap flow rather than water-use. Could be helpful since many readers will not know what the heat ratio method measures. • Water-use was changed to sap flow except where up-scaled results were discussed.

2. P1, L41: A hydrological gain could describe an increase in ET. The term is too general, better to say a gain in streamflow? • Changed to streamflow.

3. P2, L9: Hybrid genotypes are most common in Eucalyptus plantations, but they are mostly not invasive. • The word hybrid was removed.

4. P2, L20. This sentence is far too long and clumsy. • This sentence was re-worded and split into two sentences.

5. P2, 42: Surely temporal flow patterns as well? • Temporal was included.

6. P2, L46: Buffeljags River and river seen in the text. Be consistent. • Corrected throughout.

7. P3, L11: ...which is characterized by .... • Corrected.

8. P3, L19: spelling. Occurs • Corrected.

9. P3, L22: Podocarpus changed to Afrocarpus? • Corrected: Afrocarpus (Podocarpus falcatus).

10. P3, L34. Strictly speaking, Plant area index, since light interception is also by twigs, branches, stems. • Agreed but left unchanged to avoid confusion.
11. P3, L40: Vepris. After first mention, abbreviate to V. lanceolata. • Corrected.

12. P3, L42. Spelling. Diameter • Corrected.

13. P4, L8: insert comma • Corrected.

14. P4, L32. Check format of units throughout MS. Inconsistent. • Corrected.

15. P4, L40: ...dendrometer, and canopy height... • Corrected.

16. P4, L41: I think it would be clearer to refer to aerodynamic fetch rather than reach? • Corrected.

17. P4, L44: Spelling. Indigenous • Corrected.

18. P4, L46: Is this stem density sph or the wood density of stems? • Clarified, stem density.

19. P4, L47: Not sure if it is worth introducing the term medoid to the readers! • Discussed but left unchanged as it is statistically the correct term rather than saying representative, which could be subjective if not statistical.

20. P5, L6: Spelling again. Indigenous • Corrected.

21. P5, L19: Were observation borehole measurements occasional or continuous. • Due to no solinist level loggers being available for this project, only occasional borehole measurements (with a dip meter) were possible.

22. P5, L29. Spelling. significantly. • Corrected.

23. P5, L40: ...occasional maximum temperatures exceeded 40°C ? • Corrected. The word 'occasional' was used to describe days when the temperature exceeded this value. This was to avoid the sentence suggesting the everyday exceeded this temperature.

24. P6, L33: After first mention, abbreviate to C. africana • Corrected.

25. P7, L3: ...very low, dropping ... • Corrected.

26. P7, L7: ..and a porosity of 0.42... • Corrected.

27. P7, L9: 22 hours to get to FC or wilting point? Seems very fast, even for a sand. • Sentence re-worded. "The drying curve, after an isolated event, took on average 22 hours from saturation to the expected field capacity"

28. P7, L10: Spelling. Greater • Corrected.

29. P7, L18: Throughfall and stem flow are surely not interception storage? • Corrected

30. P7, L20: ...indigenous site took much longer ... No comma • Corrected.

31. P7, L38: ...water use... Be consistent about using a hyphen. • Corrected.

32. P8, L4: C. africana • Corrected.

33. P8, L28: aerodynamic fetch rather than reach? • Corrected as before.

34. P8, L34: ..a stand of introduced A. mearnsii... Sounds better than an introduced stand? • Corrected.

35. P8, L34: ..can annually use up to six .... • Corrected but said: 'can use up to six times more water annually. . .'

36. P8, L37: But see Cullin Gorgens and Marais paper which offers some quantified estimates. • Updated comparisons

37. P10, L1 onwards: Found 16 formatting errors! Please check carefully. • Corrected.

38. Table 1. Suggest be consistent in arranging trees from small to large. The first two groups of sample trees have a different sorting order. Moisture fraction is very variable in group 2 (A. mearnsii). Do you know why? • Tree sizes re-ordered. The variation in moisture content was possibly due to the different ages and sizes of the

**HESSD**

trees measured (variations in sap wood depth and active xylem concentration), which may explain these differences. This was included on page 4, line 20.

39. Figure 2. Numbers within bars along the X axis are partly obscured. • Corrected.

40. Figure 3. Caption says solar radiant density. Y2 axis label says solar radiation. Be consistent. Format of units again variable. The point separating each unit is sometimes in a lower or middle position. Check throughout the manuscript. • Corrected.

41. Figure 4: ...lower reach alien stand... Consider referring consistently to A. mearnsii rather than alien stands. • Corrected.

42. Figure 7. Pity no water table trends shown! • Due to little change in the infrequent measurements, these were not included as they did not provide further insight.

43. Table 2. Thousands separated by blank spaces here, but not consistently through the paper. Check throughout. • Corrected.

44. Figure 8. I think Et in the legend should be Et0 ? Again, introduced stand might be better described as A. mearnsii? • Corrected.

Please also note the supplement to this comment:
http://www.hydrol-earth-syst-sci-discuss.net/hess-2016-650/hess-2016-650-AC1-supplement.pdf

**Supplement:**

[revised manuscript text omitted]

---

## Author Comment (AC2) · 28 Mar 2017

RC1: 'Water-use Dynamics of an Alien Invaded Riparian Forest within the Mediterranean Climate Zone of the Western Cape, South Africa', Anonymous Referee #1, 20 Feb 2017

HESS-2016-650

Anonymous referee #1 (AR1) is thanked for their thorough review. The comments and suggestions provided were insightful and beneficial to the progress of this paper.

1. AR1 stated that there was a lack of international literature cited in the paper. "Invasive alien plants are a global problem and the authors should present their manuscript

in a way that appeals to an international audience." • This comment was well received. A further 12 citations from relevant international studies was included in the introduction, and were later compared to the findings discussed in this paper. Although some global problems differ to the South African context, the inclusion of these findings is still relevant as suggested. • An example of this is as follows: "invasive plants use up to 136 % more water than the indigenous species at the leaf scale (Baruch and Fernandez, 1993; Dixon et al., 2004; Pratt and Black, 2006). At the plant scale there is a diverse range in water-use ranging from the invasive species using 100 % less to 150 – 300 % more water than the indigenous species (Cleverly et al., 1997; Nagler et al., 2003; Kagawa et al., 2009). At the ecosystem scale studies indicate that invasive species use 189 % more water than indigenous dominates areas, particularly in tropical moist forests (Nosetto et al., 2005; Yepez et al., 2005; Fritzsche et al., 2006)."

2. AR1 suggested that the authors use their data to validate a relevant international model, which would further assist the paper in comparing their findings to findings highlighted in international literature. • Although a good suggestion, the authors decided to exclude model results in this paper. This was partly due to a second paper that will be submitted subsequent to the publication of this paper, which uses the data from this study to validate an international model (SWAT). Recommendations were included in the discussion suggesting that this data be used to calibrate and validate hydrological models.

3. AR1 stated that the 'Discussion and Conclusion' section was very weak and needed to be revised. • The authors agreed with this comment. In order to address this comment, this section was revised. Reference was made to international findings and how these compared to the results provided in the paper. Further information was provided on the importance of the findings and how they could be used for follow-up rehabilitation and research.

4. AR1 queried how the authors obtained total evaporation measurements when only tree transpiration was measured. • A closed canopy was assumed as there was

little to no short understorey vegetation, only litter. This assumption has been included in Section 3.4. Evaporation from the litter and canopy was assumed to be minimal. However, total evaporation was changed to transpiration as the authors agree that this is scientifically correct.

Specific Comments

1. Abstract line 26: add the phrase "of the heat pulse velocity sap flow technique" after the "heat ratio method". • Changed. 2. Abstract general: presenting the water use values in litres only doesn't mean much to a water resources manager or other decision maker. I would prefer to see area averaged transpiration values (in mm). • Both litres and mm were used for the benefit of readers specialized in this field as well as to the more general reader, such as a decision maker. This was updated in the abstract.

3. Pg 3 lines 32-44: How was the LAI measured? • A description of the Li-Cor (LAI-2200) plant canopy analyser was provided in the methods.

4. Pg 3 line 48: Indicate that the diameter was measured at breast height (DBH) • Changed.

5. The HRM method of the HPV technique is well known in eco-physiological/eco-hydrological cycles. So there is no need for the details on Pg 4 lines 15 to 35. Remove this text. • Although the authors considered this, they decided to leave the methods used for the HPV technique as some of the readers are not familiar with the technical component used in this study and it would further benefit students using this technique elsewhere.

6. What would add value is more detail on: probe insertion depths and circumferential arrangements and how the conducting sap wood area was determined. • The authors agreed that this would be useful information to include. This has been updated in the methods section.

7. Pg 4 line 43 – sentence beginning with " Given the homogenous: : :." Doesn't make sense. Please revise. • Sentence revised: "Due the homogenous composition of the alien stand and the dominance of Vepris and Celtis species within the indigenous stand. . ."

8. Pg 5 line 20: To confirm direct use of ground water, it would have been nice to do this using stable isotopes rather. Include this data if you have it. • Unfortunately due to budget constraints, this was not available.

9. Pg 6 first paragraph: Fig 8 is discussed before Figs 6 & 7 which only appear on pg 7. Either revise the text or change the order of the Figs. There is also a mismatch with the units. • Both the figure numbering and units were corrected.

10. The text, Pg 7 line 7 presents L/d, but Fig 8 presents water use in mm. Clarify this. • Reference to Figure 8 was included in Section 3.4 where upscaled results were discussed. The units of L day-1 were used for single tree results (allowing for tree comparisons) whereas mm was used for upscaled results that could be compared to water balance components.

11. Pg 7, lines 39-40: Not clear if the annual transpiration and ETo are for a single year or averages over 3 years. Please clarify. • As the MAP was consistently high over the three years, with little variation between years, monthly averages over the three year monitoring period were used. This was done to simplify the graphs.

12. Pg 8 "Discussion and Conclusion". As stated earlier, this section is very weak and I would prefer to see; 1) comparisons between the results found here and what other researchers found, and; 2) comparisons with international literature. • As discussed in the general comments, this section has been revised.

13. I also have issues with the references: o The following references are missing in the reference list 1. Everson et al., 2007; 2. Richardson et al., 2007; 3 O'Grady et al., 2005; 4. Miller et al., 2007; 5; Everson et al., 2006; 6. Uddin and Smith, 2014. Format

of the references in the references list is not consistent. Some journals are written in full while others are abbreviated. In places it is not clear to who reports are written for e.g. pg 11 lines 10-11 and other places. For other references, journal names are replaced with address of the institution. For example, pg 11 line 38 is a publication in the Agricultural and Forest Meteorology journal which is bizarrely replaced by the institutional address. • The references were corrected

Please also note the supplement to this comment:
http://www.hydrol-earth-syst-sci-discuss.net/hess-2016-650/hess-2016-650-AC2-supplement.pdf

---

## Author Response (AR2)

**Editor: 'Water-use Dynamics of an Alien Invaded Riparian Forest within the Mediterranean Climate Zone of the Western Cape, South Africa', 6 May 2017**

**HESS-2016-650**

The editor is thanked for the comprehensive review. The comments and suggestions provided were appreciated by the authors and used to improve the paper.

1. **The editor stated that the authors should be aware that rainfall and evaporation are fluxes, and therefore when values are given for each one of these, the time interval over which each one of these was measured or estimated must be included. Notes were made throughout the paper. A correction was applied to all units throughout the paper (where relevant). An example of this was:**

   - 636 mm was changed to 636 mm·$a^{-1}$ ($a^{-1}$ was used throughout instead of $yr^{-1}$)

2. **The editor stated that authors should be aware that words such as less, more, smaller, higher are comparative words. When writing the two items being compared must be stated.**

   - The authors apologize for these errors. These have since been corrected. If comparative differences were made, both variables were mentioned. Otherwise, the comparative word was changed.

3. **The authors were told to check Figure 2 and make sure that rainfall axis is correctly written "mm/month". This was corrected in the figure and elsewhere.**

4. **The author stated: "Figure 3 is very confusing. The left Y axis has negative temperature values of up to -40 degrees. The right Y axis has rainfall and radiation combined. Which values relate to the top graph, and the bottom graph?"**

   - The authors corrected this figure. Negative values were removed and legend labels were moved to avoid confusion.

5. **In Figure 4 and 5, what does each one of the lines represent, i.e. the dashed line and black line with markers? Readers are not expected to guess what a line in a diagram represents. Include a legend.**

   - A legend has since been included in these figures.
   - Reference was made to 'indigenous' and 'alien invasive' trees in the captions.

6. **In Figure 6 and 7, are you referring to soil water content on the Y axis and caption? What does "corresponding" in the caption mean?**

   - Reference was made to 'soil water content'. This was also applied in the text throughout the document.
   - The word corresponding was removed, stating only the variables displayed in the graph.

7. **The author stated that Table 2 is confusing. In the column labelled "Species" you have in each row the same name of a species written 3 times. Are these different? Presumably this could be referring to "small", "medium" and "large" trees. If that is the case, state this clearly. A reader should be able to understand information in a table without guessing. If the flow rate is per day, then clearly write this in the column headings, e.g. L/day, L/year.**

- The tables were explained in text. This will make sense once the table is positioned in the document.
- The units for flow rate per day were corrected in the Table.

8. **In Figure 8 the label for y axis, should be "mm/month". What does Et mean? Why is it that the Y axis has no continuous line? Avoid adopting Excel default formats which are not acceptable in publications.**

- The y-axis label was corrected.
- Et was written out.
- The graph was changed to a continuous line graph.

In text comments were applied throughout the documents. All of these were addressed by the authors.

---

## Author Response (AR3)

**Editor: 'Water-use Dynamics of an Alien Invaded Riparian Forest within the Mediterranean Climate Zone of the Western Cape, South Africa', 16 May 2017**

**HESS-2016-650**

The editor is thanked again for the comprehensive review. The amendments provided here are in response to the final editor's comments. The manuscript has been amended and submitted separately.

1. **The editor stated that the revised manuscript has greatly improved. The authors should provide the units of measurement for Vh, v1 and v2 in Equation (1):**

   - The units were included in the text description of the equation. $V_h$ - cm·hr$^{-1}$ and v1 and v2 - °C.

2. **In Table 1, the columns presenting the diameter and class size of diameters should be given first before wood density so that readers will understand the differences between rows with the same species names. Please provide below the table a note showing the meaning of S, M, L, given in the class size column:**

   - Table 1 was rearranged to include diameter and size class first. S, M and L was changed to small, medium and large respectively.

3. **In Table 2 I assume that different rows with the same species name refer to different diameter classes. A column showing this information should be included, otherwise you confuse the readers**

   - Table 2 was amended to include diameter classes.